# ER assembly of SNARE complexes mediating formation of partitioning membrane in *Arabidopsis* cytokinesis

**Matthias Karnahl[1†], Misoon Park[1†], Ulrike Mayer[2], Ulrike Hiller[1,2], Gerd Jürgens[1*]**

[1]Center for Plant Molecular Biology (ZMBP), Developmental Genetics, University of Tübingen, Tübingen, Germany; [2]Center for Plant Molecular Biology (ZMBP), Microscopy, University of Tübingen, Tübingen, Germany

**Abstract** Intracellular membrane fusion mediates diverse processes including cell growth, division and communication. Fusion involves complex formation between SNARE proteins anchored to adjacent membranes. How and in what form interacting SNARE proteins reach their sites of action is virtually unknown. We have addressed this problem in the context of plant cell division in which a large number of TGN-derived membrane vesicles fuse with one another to form the partitioning membrane. Blocking vesicle formation at the TGN revealed *cis*-SNARE complexes. These inactive cytokinetic SNARE complexes were already assembled at the endoplasmic reticulum and, after passage through Golgi/TGN to the cell division plane, transformed into fusogenic SNARE complexes. This mode of trafficking might ensure delivery of large stoichiometric quantities of SNARE proteins required for forming the partitioning membrane in the narrow time frame of plant cytokinesis. Such long-distance trafficking of inactive SNARE complexes would also facilitate directional growth processes during cell differentiation.

**\*For correspondence:** gerd. juergens@zmbp.uni-tuebingen.de

[†]These authors contributed equally to this work

**Competing interests:** The authors declare that no competing interests exist.

## Introduction

Cytokinesis partitions the cytoplasm of the dividing cell. In non-plant eukaryotes, a contractile acto-myosin ring strongly reduces the area of contact between the forming daughter cells. Consequently, little membrane expansion has to be supported by membrane traffic and fusion, which can largely be afforded by local recycling (*Nakayama, 2016*). In contrast, dividing plant cells lack the contractile actomyosin ring and thus have to make a large partitioning membrane – named cell plate – which is progressively formed from the centre to the periphery of the cell (*Müller and Jürgens, 2016*). The cell plate originates from membrane vesicles that fuse with one another upon delivery to the plane of cell division along the highly dynamic microtubules of the phragmoplast. Membrane fusion during *Arabidopsis* cytokinesis requires a cytokinesis-specific Qa-SNARE (aka syntaxin) named KNOLLE (*Lauber et al., 1997*), which forms two distinct functionally overlapping SNARE complexes by interaction with different sets of promiscuous SNARE partners: (i) QaQbcR-complex containing Qbc-SNARE SNAP33 and R-SNARE VAMP721 or VAMP722, and (ii) QaQbQcR-complex containing Qb-SNARE NPSN11, Qc-SNARE SYP71 and R-SNARE VAMP721 or VAMP722 (*El Kasmi et al., 2013*; *Heese et al., 2001*; *Zheng et al., 2002*). The formation of these *trans*-SNARE complexes requires the cytokinesis-specific action of the Sec1/Munc18 (SM) protein KEULE, which interacts with mono-meric KNOLLE but not with the assembled KNOLLE-containing SNARE complex at the plane of cell division (*Park et al., 2012*). It is not known in what form the SNARE proteins reside on the vesicles prior to the action of the SM protein. One possibility is that each SNARE protein is trafficked sepa-rately and kept in its monomeric form until fusion. Alternatively, one or more SNARE proteins might form inactive complexes. Or there might be a mixture of vesicles, some bearing the R-SNARE and

others a preassembled Q-SNARE complex. Here, we examine in what form – monomers or complexes – SNARE proteins are present on the cytokinetic vesicles and where along the trafficking pathway complexes of cytokinetic SNARE proteins might be formed.

## Results and discussion

The cytokinesis-specific Qa-SNARE KNOLLE is made during late G2/M phase and turned over rapidly at the end of cytokinesis (*Lauber et al., 1997*). Newly synthesised KNOLLE protein is inserted into the membrane of the endoplasmic reticulum (ER) and traffics along the secretory pathway via Golgi stack and *trans*-Golgi network (TGN) to the plane of cell division (*Figure 1A*) (*Reichardt et al., 2007*). Upon cell-plate formation, KNOLLE is endocytosed and targeted via multivesicular body (MVB) to the vacuole for degradation (*Figure 1A*) (*Reichardt et al., 2007*). Unlike the situation in mammals, yeast and most flowering plants, secretory traffic in *Arabidopsis* is insensitive to the fungal toxin brefeldin A (BFA). BFA inhibits the ARF-activating guanine-nucleotide exchange reaction of sensitive ARF-GEFs, thus preventing the formation of transport vesicles (*Mossessova et al., 2003*; *Renault et al., 2003*). We have engineered in *Arabidopsis* a BFA-inducible system with which secretory traffic can be blocked at two specific sites along the route, ER and TGN. Relevant BFA-insensitive ARF-GEFs, human GBF1-related GNL1 or human BIG1-related BIG3, were eliminated by mutation, leaving BFA-sensitive GNOM and BIG1,2,4, respectively (*Richter et al., 2007*, *2014*). Consequently, BFA treatment of *gnl1* mutant plants prevents recruitment of COPI coat complexes to the *cis*-Golgi membrane, causing collapse of the ER-Golgi traffic. Qa-SNARE KNOLLE is thus retained in the ER and cytokinesis is impaired, resulting in binucleate cells (*Figure 1A*) (*Richter et al., 2007*). Late-secretory traffic from the TGN to the plane of cell division requires the formation of AP-1 complex-coated transport vesicles, which depends on the action of four functionally overlapping ARF-GEFs BIG1 to BIG4 (*Park et al., 2013*; *Richter et al., 2014*). Mutational inactivation of the sole BFA-resistant ARF-GEF BIG3 renders AP-1 vesicle formation BFA-sensitive. KNOLLE is thus retained at TGN membrane aggregates called BFA compartments and cytokinesis is impaired, resulting in binucleate cells (*Figure 1A*) (*Richter et al., 2014*). To examine in what form – monomeric or part of complex – KNOLLE is delivered to the plane of cell division, we inhibited secretory traffic by BFA treatment of *gnl1* and *big3* mutants. However, BFA treatment would inhibit secretory traffic in some dividing cells but not in others because the cells in the developing seedling root divide asynchronously. To overcome this limitation, we used β-estradiol (EST)-inducible expression of KNOLLE SNARE partners NPSN11 or SNAP33 fused to a fluorescent protein (*Zuo et al., 2000*). Importantly, neither YFP:NPSN11 nor GFP:SNAP33 was expressed without EST treatment (*Figure 1B–C*), which is a prerequisite for the detection of newly-made cytokinesis-specific SNARE complexes.

Arabidopsis *big3* mutant seedlings were treated with BFA for 30 min followed by 210 min of combined BFA and EST treatment to induce expression of YFP:NPSN11 or GFP:SNAP33 in cells whose traffic to the cell-division plane was blocked at the TGN. Seedlings were then live-imaged for YFP: NPSN11 or GFP:SNAP33. The two fusion proteins accumulated in TGN-containing BFA compartments; this was in contrast to the strong labeling of cell plates in wild-type seedlings expressing BFA-resistant BIG3 ARF-GEF or in *big3* mutant seedlings not treated with BFA (*Figure 1B–C*). The frequency of cells undergoing cytokinesis was not altered by BFA treatment of wild-type or mutant seedlings, as evidenced by immunostaining of phragmoplast microtubules (*Figure 1—figure supplement 1*; *Supplementary file 1a*).

Co-immunoprecipitation analysis of BFA-treated *big3* mutant seedlings expressing EST-inducible YFP:NPSN11 revealed the presence of Qb-SNARE NPSN11 fused to YFP, Qa-SNARE KNOLLE, Qc-SNARE SYP71 and R-SNARE VAMP721/722 as well as the absence of Qbc-SNARE SNAP33 in the anti-GFP precipitate (*Figure 2A*). Thus, only the members of the KNOLLE-NPSN11-SYP71-VAMP721/722 complex were co-immunoprecipitated whereas the SNARE partner SNAP33 from the other KNOLLE-containing SNARE complex was not. The converse was observed in the co-immunoprecipitation analysis of *big3* mutant seedlings expressing the EST-inducible Qbc-SNARE member of the other KNOLLE complex, GFP:SNAP33. Qc-SNARE SYP71 was not detected in the co-immunoprecipitate, in contrast to the members of the trimeric KNOLLE complex Qbc-SNARE GFP: SNAP33, Qa-SNARE KNOLLE and R-SNARE VAMP721/722 (*Figure 2B*). Thus, the interaction detected by co-immunoprecipitation was exclusively confined to members of the KNOLLE-containing complex that contained the EST-induced SNARE partner, strongly suggesting that only direct

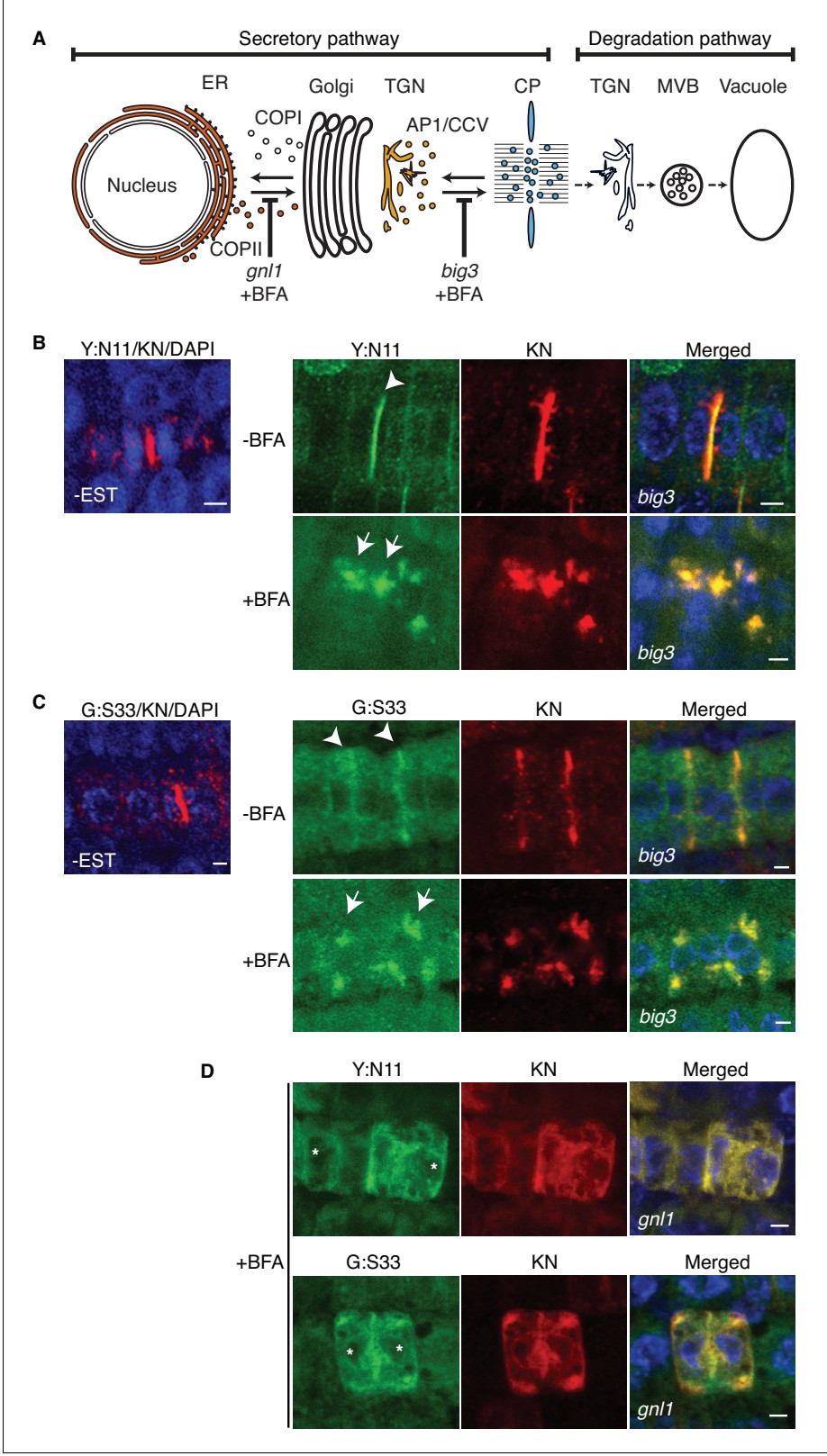

**Figure 1.** Site-specific inhibition of SNARE protein trafficking to the cell-division plane. (**A**) Qa-SNARE KNOLLE trafficking route in cytokinesis (*Reichardt et al., 2007*). ER, endoplasmic reticulum; TGN, *trans*-Golgi network; CP, cell plate; MVB, multivesicular body; COPI, COPII, AP1/CCV, membrane vesicles with specific coat protein complexes; *gnl1*, *big3*, knockout mutations of ARF-GEFs rendering those trafficking steps sensitive to brefeldin A

*Figure 1 continued on next page*

*Figure 1 continued*

(BFA). (**B–D**) Subcellular localisation of estradiol-inducible YFP:NPSN11 (**B, D**; green) and GFP-SNAP33 (**C, D**; green), and KNOLLE (**B–D**; red) in *big3* (**B, C**) and *gnl1 GNL1*<sup>BFA-sens.</sup> (**D**) mutant seedling roots treated with 50 μM BFA for 30 min, followed by 50 μM BFA + 20 μM estradiol for 210 min. Note that YFP:NPSN11 (Y:N11) or GFP: SNAP33 (G:S33) accumulates with KNOLLE (KN) at the BFA compartments in BFA-treated *big3* mutant whereas YFP:NPSN11 or GFP:SNAP33 colocalises with KNOLLE at the ER in BFA-treated *gnl1 GNL1*<sup>BFA-sens.</sup> mutant. Note also no expression of YFP:NPSN11 (Y:N11, **B**) or GFP:SNAP33 (G:S33, **C**) without estradiol treatment. Nuclei of overlays (**B–D**) were counterstained with DAPI (blue). -BFA, mock treatment; +BFA, BFA treatment; -EST, no estradiol treatment. Arrowheads, cell plates; arrows, BFA compartments; asterisks, ER. Scale bar, 5 μm. The experiments were technically repeated three times.

The following figure supplements are available for figure 1:

**Figure supplement 1.** Cytokinetic cells in *big3* and *gnl1 GNL1*<sup>BFA-sens.</sup> mutant seedling roots.

**Figure supplement 2.** Subcellular localisation (**A, B**) and co-immunoprecipitation analysis (**C**) of pKNOLLE::mRFP: PEP12 (aka SYP21) (red) in *big3* mutant seedling root cells expressing estradiol-inducible YFP:NPSN11 (**A, C**) and GFP:SNAP33 (**B, C**).

**Figure supplement 3.** Site-specific inhibition of SNARE protein trafficking to the cell-division plane and loss of COPI from Golgi membrane in *gnl1 GNL1*<sup>BFA-sens.</sup> seedlings.

---

interactions between SNARE complex members were detected. Further co-immunoprecipitation experiments with non-EST-induced seedlings demonstrated that co-immunoprecipitation of KNOLLE indeed required the EST-induced expression of YFP:NPSN11 or GFP:SNAP33 (*Figure 2—figure supplement 1A*). In addition, to rule out that these complexes might have formed during the immunoprecipitation procedure, we mixed the protein extracts with varying amounts of extract from *KNOLLE::mCherry:KNOLLE* transgenic seedlings before immunoprecipitation with anti-GFP beads. Neither 1x nor 10x mCherry:KNOLLE addition changed the amount of KNOLLE-containing SNARE complex formed (*Figure 2—figure supplement 1B*). We also addressed whether BFA treatment might stimulate the formation of KNOLLE-containing SNARE complexes. To this end, we compared protein extracts from untreated wild-type seedlings with those from BFA-treated wild-type seedlings. No obvious difference in KNOLLE complex formation was detected between treated and untreated seedlings (*Figure 2—figure supplement 1C*). As an additional control, we examined whether EST-induced KNOLLE-partners GFP:SNAP33 and YFP:NPSN11 co-immunoprecipitated the Qa-SNARE PEP12 (aka SYP21). PEP12 is normally located at the multivesicular body (MVB) (*da Silva Conceição et al., 1997*; *Müller et al., 2003*), and was also relocated to the same BFA compartments as was YFP:NPSN11 in BFA-treated *big3* mutant seedlings (*Figure 1—figure supplement 2A–B*). Neither SNAP33 nor NPSN11 interacted with mRFP-tagged PEP12 whereas both did interact with KNOLLE, confirming the specificity of the co-immunoprecipitation assay (*Figure 1—figure supplement 2C*). These results indicate that KNOLLE forms part of a SNARE complex before the initiation of the fusion process at the plane of cell division. Thus, KNOLLE seems to be transported as part of two different *cis*-SNARE complexes from the TGN to the plane of cell division. These assembled SNARE complexes comprise either (i) KNOLLE, SNAP33 and VAMP721/722 or (ii) KNOLLE, NPSN11, SYP71 and VAMP721/722.

To determine where along the secretory pathway the KNOLLE-containing *cis*-SNARE complexes are assembled, we blocked traffic already at the ER-Golgi interface by BFA treatment of *gnl1* mutant seedlings expressing engineered BFA-sensitive GNL1<sup>BFA-sens.,</sup> (*Richter et al., 2007*) and EST-inducible SNAREs YFP:NPSN11 or GFP:SNAP33. By subcellular localisation, all relevant SNARE components (YFP:NPSN11, GFP:SNAP33, KNOLLE) were detected at the ER (*Figure 1D*), indicating effective inhibition of traffic between ER and Golgi stacks. As an additional control, we analysed the subcellular localisation of COPI subunit γCOP, which is normally associated with the Golgi membrane whereas BFA treatment caused accumulation of γCOP in the cytosol (*Figure 1—figure supplement 3*) (*Richter et al., 2007*). Co-immunoprecipitation with anti-GFP beads of protein extracts from BFA-treated BFA-sensitive *GNL1*<sup>BFA-sens.</sup> seedlings revealed that KNOLLE already exists as part

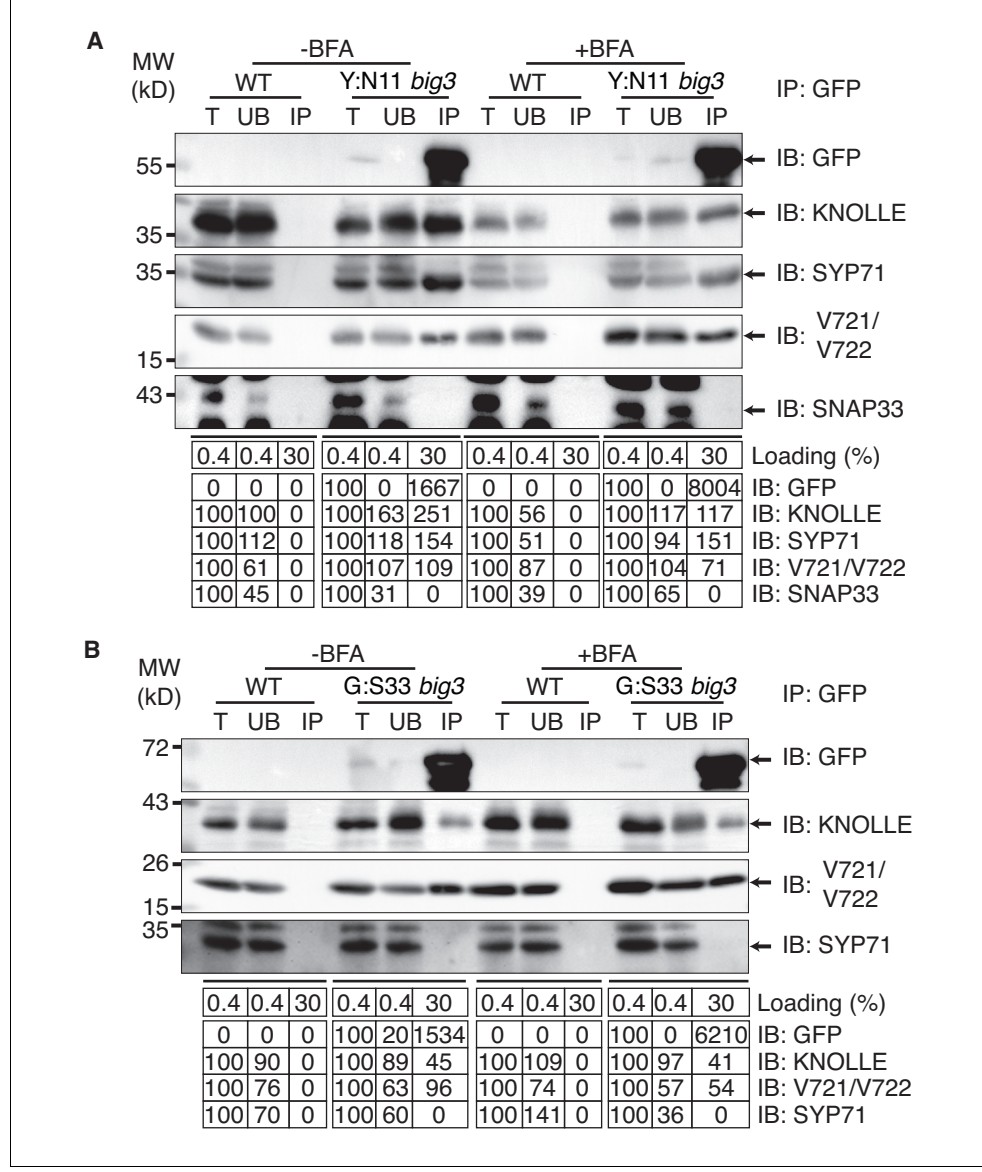

**Figure 2.** Interaction analysis of cytokinetic SNAREs with traffic blocked at the TGN. Wild-type (WT) and *big3* mutant seedlings carrying estradiol-inducible *YFP:NPSN11* (**A**) or *GFP:SNAP33* (**B**) transgenes were treated with 50 µM BFA for 30 min followed by 50 µM BFA + 20 µM estradiol for 210 min (see *Figure 1B–C*). Protein extracts were subjected to immunoprecipitation with anti-GFP beads, protein blots were probed with the antisera indicated on the right (IB): GFP, anti-GFP; KN, anti-KNOLLE; V721/V722, anti-VAMP721/722; SYP71, anti-SYP71; SNAP33, anti-SNAP33; kDa, protein size (left); MW, molecular weight; -BFA, mock treatment; +BFA, BFA treatment; T, total extract; UB, unbound; IP, immunoprecipitate. Loading (%), relative loading volume to total volume; relative signal intensity (input signal = 100% for UB and IP). The experiments were technically repeated more than six times.

The following figure supplement is available for figure 2:

**Figure supplement 1.** Control experiments for co-immunoprecipitation analysis of cytokinetic SNAREs.

of a *cis*-SNARE complex in the ER. Further co-immunoprecipitation analysis demonstrated interactions exclusively between components within each cytokinetic SNARE complex but not between members of the two different SNARE complexes, ruling out recovery of non-interacting proteins from the same membrane compartment (*Figure 3A–B*). Thus, KNOLLE is trafficked in two different *cis*-SNARE complexes along the secretory pathway from the ER to the plane of cell division.

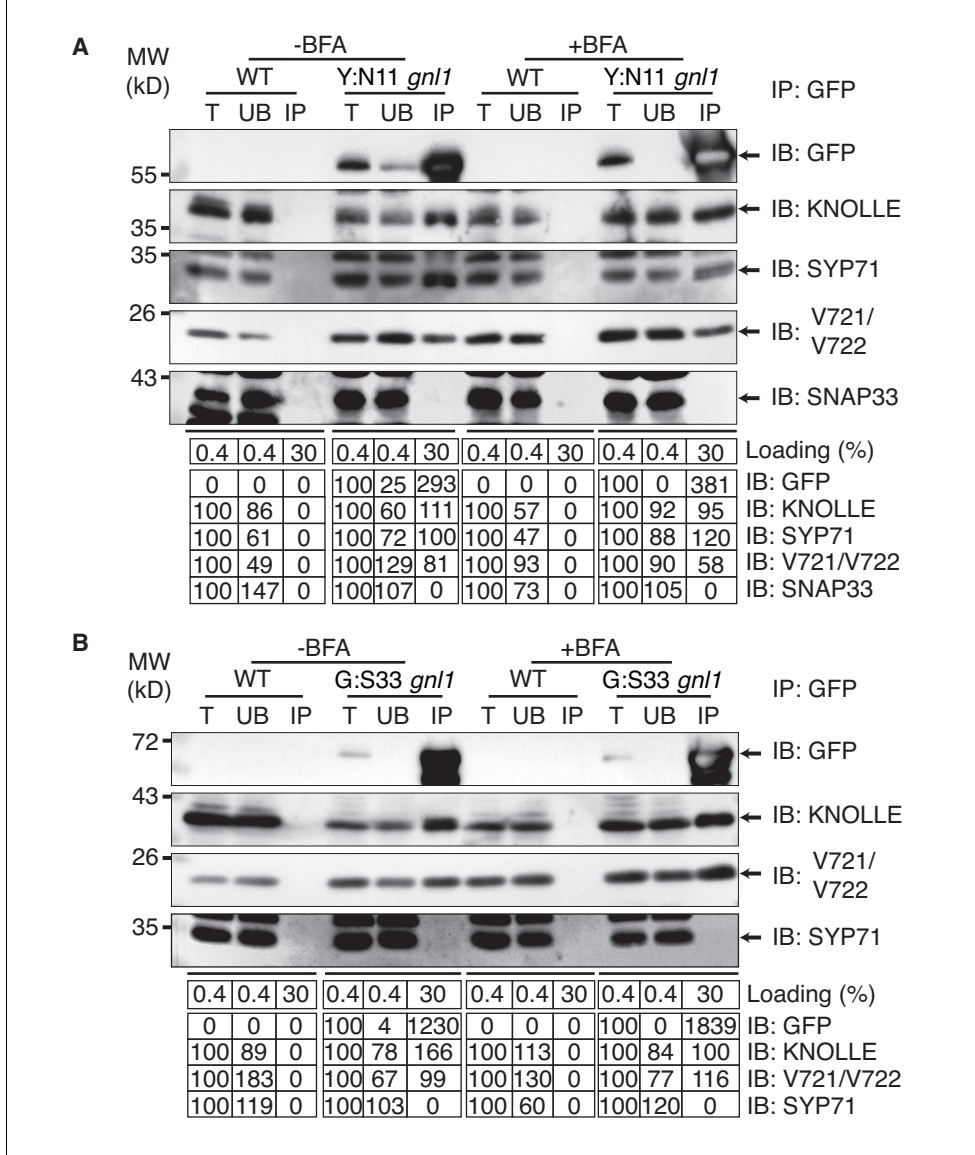

**Figure 3.** Interaction analysis of cytokinetic SNAREs with traffic blocked at the ER. Wild-type (WT) and *gnl1* mutant seedlings complemented with *GNL1^BFA-sens.^* encoding a BFA-sensitive variant of GNL1 and carrying estradiol-inducible *YFP:NPSN11* (**A**) or *GFP:SNAP33* (**B**) transgenes were treated with 50 μM BFA for 30 min followed by 50 μM BFA + 20 μM estradiol for 210 min (see *Figure 1D*). Protein extracts were subjected to immunoprecipitation with anti-GFP beads, protein blots were probed with the antisera indicated on the right (IB): GFP, anti-GFP; KN, anti-KNOLLE; V721/V722, anti-VAMP721/722; SYP71, anti-SYP71; SNAP33, anti-SNAP33; kDa, protein size (left); MW, molecular weight; -BFA, mock treatment; +BFA, BFA treatment; T, total extract; UB, unbound; IP, immunoprecipitate. Loading (%), relative loading volume to total volume; relative signal intensity (input signal = 100% for UB and IP). The experiments were technically repeated more than six times.

Our results indicate that cytokinetic SNARE complexes are assembled on the ER and from there delivered as *cis*-SNARE complexes rather than monomeric SNARE proteins along the secretory pathway, via Golgi stack and TGN, to the plane of cell division (*Figure 4*). This implies that *cis*-SNARE complexes (i.e. residing on the same membrane) are delivered to the division plane where they are transformed into fusogenic *trans*-SNARE complexes linking adjacent membrane vesicles. These observations also explain the requirement for the SM protein KEULE during cell-plate formation (*Park et al., 2012*). Breaking up *cis*-SNARE complexes by the action of NSF ATPase, which normally

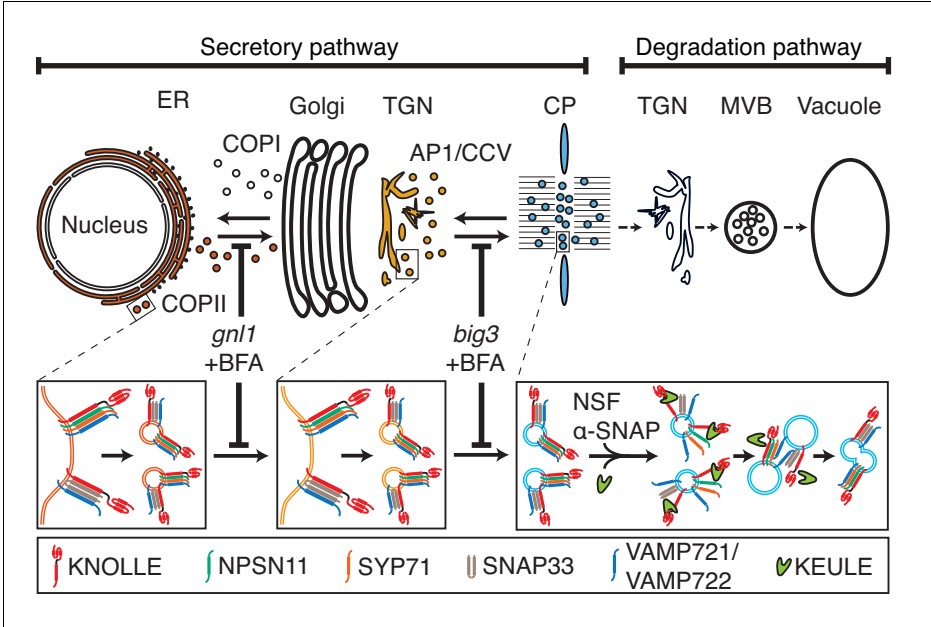

**Figure 4.** Trafficking of *cis*-SNARE complexes during cytokinesis (model). Two different types of cytokinetic *cis*-SNARE complexes are assembled on the ER, recruited into COPII vesicles and passed on to the Golgi stack/TGN. At the TGN, they are incorporated into AP1/CCV vesicles for delivery to the division plane. Following their disassembly by NSF ATPase, monomeric Qa-SNARE KNOLLE is assisted by SM protein KEULE in the formation of *trans*-SNARE complexes mediating fusion of adjacent vesicles during cell-plate formation and expansion (*Park et al., 2012*).

occurs following the fusion of membrane vesicles with the target membrane (*Rizo and Südhof, 2012*) would result in back-folding of the monomeric Qa-SNARE to prevent re-formation of the *cis*-SNARE complex. In cytokinesis, however, SM protein KEULE interacts with Qa-SNARE KNOLLE to keep it open as a prerequisite for the formation of the fusogenic *trans*-SNARE complex (*Park et al., 2012*).

Trafficking of *cis*-SNARE complexes has two major advantages: (i) the *cis*-SNARE complex is an energetically favoured inactive form (*Jahn et al., 2003*) that is well-suited for transport and also ensures equal amounts of SNARE partners being delivered to the site of action. The four-helical bundle of the SNARE domains is very stable, requiring ATP hydrolysis for its disassembly. Because of its stability, the assembled *cis*-SNARE complex is physiologically inactive, not interacting with other SNARE proteins. (ii) Moreover, this might be a highly economic strategy of meeting the sharply rising demand for membrane-fusion capacity during cytokinesis when the equivalent of about one-third of the cell surface area has to be produced in the plane of cell division in a narrow time frame of about 30 min. In animal cytokinesis, the problem is largely solved by reducing the surface area through the constriction by the contractile ring that pulls in the plasma membrane. The remaining gap in the centre of the division plane is then closed by vesicle fusion that is mediated by plasma membrane SNARE proteins present throughout the cell cycle (*Low et al., 2003*) and/or by ESCRTIII activity (*Mierzwa and Gerlich, 2014*). Apart from plant cytokinesis, any major expansion of the eukaryotic cell surface area requires enhanced membrane fusion capacity that cannot easily be matched by the local recycling of plasma membrane-resident SNARE proteins whereas the long-distance delivery of inactive *cis*-SNARE complexes proposed here would meet the requirement.

## Materials and methods

### Plant material, Growth Condition and Transformation

*Arabidopsis thaliana* [NCBITaxon:3702] wild-type (Columbia, Col), *pKNOLLE::mCherry:KNOLLE* or mutant plants were grown on soil or media (1/2 MS medium, 0.1% MES, pH 5.6) at 23°C in continuous light condition. *big3* homozygous plants were transformed with *pMDC7::GFP:SNAP33* or *pMDC7::YFP:NPSN11* using *Agrobacterium* [NCBITaxon:358]-mediated floral dipping (*Clough and Bent, 1998*; *Richter et al., 2014*). T1 seedlings were selected on Hygromycin (20 µg/ml, Duchefa Biochemie, Netherlands) plates to isolate *big3* mutant plants carrying transgenes *pMDC7::GFP: SNAP33* or *pMDC7::YFP:NPSN11*. The same transgenes were introduced into a BFA-sensitive *GNL1* genetic background by crossing these transgenic plants with *gnl1* homozygous plants bearing a *pGNL1::GNL1^BFA-sens.* transgene (*Richter et al., 2007*). For interaction analysis of NPSN11 and SNAP33 with MVB-localised Qa-SNARE PEP12 (aka SYP21), *big3* homozygous plants bearing *pMDC7::GFP:SNAP33* or *pMDC7::YFP:NPSN11* were transformed with *pKNOLLE::mRFP:PEP12*. T1 plants were selected by spraying them three times with 1:1000 diluted BASTA (183 g/l glufosinate; AgrEvo, Düsseldorf, Germany). The homozygous background of *big3* or *gnl1 GNL1^BFA-sens.* was confirmed as previously reported (*Richter et al., 2007*, *2014*).

### Molecular biology

For generating *pMDC7::GFP:SNAP33, GFP:SNAP33* was amplified with GFP-AttB1-5 and SNAP33-AttB2-3 primers from *p35S::GFP:SNAP33* (*Park et al., 2012*). According to the manufacturer's instruction (Invitrogen, Molecular Probes), the PCR product was cloned into a modified *β*-estradiol inducible *pMDC7* vector (*Zuo et al., 2000*) in which *Ubiquitin 10* promoter replaced the original promoter (kindly provided by Niko Geldner, Univ. Lausanne). For generating *pMDC7::YFP:NPSN11*, YFP: NPSN11 was amplified by PCR with YFP-AttB1-5 and NPSN11-AttB2-3 primers from *pKNOLLE::YFP: NPSN11* (*El Kasmi et al., 2013*) and further cloned into the same *pMDC7* vector as described above. For generating *pKNOLLE::mRFP:PEP12*, *PEP12* coding sequence was amplified by PCR with PEP12-XbaI-5 and PEP12-EcoRI-3 primers. The PCR products were digested with *Xba*I and *Eco*RI (Thermo Fischer Scientific, Massachusetts, US) and cloned in-frame downstream of *mRFP* in the *KNOLLE* expression cassette (*Müller et al., 2003*). For primer sequences, see *supplementary file 1b*.

### Chemical treatment

Five-day-old seedlings grown on solid media (1/2 MS, 0.1% MES, pH 5.6, 0.9% Agar) were transferred to liquid media (1/2 MS, 0.1% MES, 1% sucrose, pH 5.6) with or without 50 µM brefeldin A (BFA, 50 mM stock solution in 1:1 DMSO/EtOH, Invitrogen). After 30 min, 20 µM *β*-estradiol (EST, 20 mM stock solution in DMSO, Sigma-Aldrich, St. Louis, US) was added, and the seedlings were then incubated for another 210 min with mild agitation.

### Co-immunoprecipitation and immunoblot analysis

Co-immunoprecipitation was slightly modified from a published protocol (*Park et al., 2012*). In brief, 1–2 g of seedlings were frozen in liquid nitrogen (N$_2$) immediately after chemical treatment. The seedlings were thoroughly grounded and the powder suspended in ice-cold buffer (50 mM Tris pH 7.5, 150 mM NaCl, 1 mM EDTA, 0.5% Triton X-100) supplemented with EDTA-free complete protease inhibitor cocktail (Roche, Basel, Swiss Confederation). Cleared protein lysate was incubated with anti-GFP beads (GFP-trap, Chromotek, Planegg-Martinsried, Germany) for 2 hr in the cold room with mild rotation. The beads were washed six times with ice-cold buffer (50 mM Tris pH 7.5, 150 mM NaCl, 1 mM EDTA, 0.2% Triton X-100) supplemented with EDTA-free complete protease inhibitors cocktail and resuspended with 2x Laemmli buffer. For *Figure 2—figure supplements 1B*, 0.3 ml of cleared protein extracts of YFP:N11 or GFP:SNAP33 were incubated with 0.3 ml or 3 ml of the cleared protein extracts of mCherry:KNOLLE for 2 hr as described above and subjected to immunoprecipitation with anti-GFP beads. For immunoblot analysis, primary antisera anti-GFP (1:1000, mouse, Roche [SCR:001326]), anti-KNOLLE (KN, 1:6000, rabbit) (*Lauber et al., 1997*), anti-VAMP721/VAMP722 (V721/V722) (1:5000, rabbit) (*Kwon et al., 2008*), anti-SNAP33 (1:5000, rabbit) (*Heese et al., 2001*), anti-SYP71 (1:4000, rabbit) (*El Kasmi et al., 2013*), anti-RFP (1:700, rat, Chromotek [RRID:AB_2336064]), anti-γCOP (aka SEC21) (1:5000, rabbit, Agrisera, Vännäs, SWEDEN

[SCR:013574]) and POD-conjugated secondary antibodies (1:5000 for anti-rabbit-POD, 1:2000 for anti-rat-POD, Sigma-Aldrich [SCR:008988]) were used. Membranes were developed using a chemiluminescence detection system (Fusion Fx7 Imager, PEQlab, Erlangen, Germany).

### Immunofluorescence analysis

After chemical treatment, seedlings were immediately fixed in 4% (w/v) paraformaldehyde for 1 hr and stored at −20°C until used for immunostaining. For immunofluorescence, primary antisera anti-KN (1:4000, rabbit) (*Lauber et al., 1997*), anti-γCOP (1:2000, rabbit, Agrisera), anti-α-tubulin (1:600, rat, Abcam, Cambridge, UK [SCR:012931]) and secondary antibodies anti-rabbit Cy3 (1:600, Dianova, Hamburg, Germany), anti-rat Cy3 (1:600, Dianova) were applied. Nuclei were stained with 1 µg/ml DAPI (1 mg/ml stock solution in $H_2O$). Samples were prepared manually or with an immuno-histochemistry system (InsituPro VSi, Intavis, Cologne, Germany). Fluorescent images were taken using a confocal laser scanning microscope (Leica SP8 for *Figure 1* and *Figure 1—figure supplements 1* and *2*; Zeiss LSM880 for *Figure 1—figure supplement 3*).

### Softwares

Sequences were analysed with CLC main workbench 6. Fluorescent images were maximally projected from Z-stack images using the Fiji ImageJ program. Images were further processed using Adobe Photoshop CS3 and Adobe Illustrator CS3. For quantifying signal intensity in immunoblot analysis, the Fiji (ImageJ, NIH) program was used.

## Acknowledgements

We thank Jennifer Saile and Sandra Richter for technical assistance, Sandra Richter and Simon Klesen for construction of *pKNOLLE::mCherry:KNOLLE*, Niko Geldner for vector, Paul Schultze-Lefert, Tony Sanderfoot and Natasha V Raikhel for antisera and Thorsten Nürnberger, Martin Bayer, Christopher Grefen and Niko Geldner for critical reading of the manuscript. Funding by the Deutsche Forschungsgemeinschaft is gratefully acknowledged.

## Additional information

### Funding

| Funder | Grant reference number | Author |
|---|---|---|
| Deutsche Forschungsge-meinschaft | Ju 179/19-1 | Gerd Jürgens |

The funders had no role in study design, data collection and interpretation, or the decision to submit the work for publication.

### Author contributions

MK, Acquisition of data, Analysis and interpretation of data; MP, Acquisition of data, Writing-review and editing, Analysis and interpretation of data, Drafting and revising the article; UM, UH, Acquisition of data; GJ, Conceptualization, Funding acquisition, Writing-original draft, Writing-review and editing, Conception and design

### Author ORCIDs

Gerd Jürgens, http://orcid.org/0000-0003-4666-8308

## Additional files

### Supplementary files

• Supplementary file 1. (a) Frequency of cytokinetic cells in mutant seedling roots. (b) Primers used in this study

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
