## [Decision Letter]

Thank you for submitting your article "ER assembly of SNARE complexes mediating formation of partitioning membrane in Arabidopsis cytokinesis" for consideration by *eLife*. Your article has been reviewed by three peer reviewers, and the evaluation has been overseen by Mohan Balsubramanian as Reviewing Editor and Christian Hardtke as the Senior Editor. The following individual involved in review of your submission has agreed to reveal his identity: Jürgen Kleine-Vehn (Reviewer #3).

The reviewers have discussed the reviews with one another and the Reviewing Editor has drafted this decision to help you prepare a revised submission.

The referees agreed that your work uncovers novel mechanisms of how SNARE proteins are held in an inactive state after assembly at the ER and potentially how these may be activated. The use of a combination of approaches including genetics, imaging, biochemistry, and inhibitor studies was also appreciated. However, a few issues have been raised, all of which should be possible to be accomplished with the tools already available in your laboratory.

Essential revisions:

1) The possibility that SNARE molecules could assemble into the complexes during procedures of immunoprecipitation should be ruled out. It would be effective to add excessive amounts of purified proteins of or plant lysates containing KNOLLE and/or VAMP721 with specific tags into the reactions to show the exogenous proteins do not affect the amounts of the SNARE complexs.

2) The immunoprecipitation experiments should be also accompanied by control using non-induced plants, Y:N11 and G:S33 plants without the EST treatment, which is useful to firmly show that authors detected only de-novo-synthesized proteins.

3) It will be interesting to know if Keule is present in any of your GFP immune complexes, given its important role in cell plate formation.

4) The entire manuscript is based on drug treatments and it remains a possibility that BFA induces certain identity changes to the compartments or cytokinetic cells. Hence, the drug itself could actually stimulate complex assembly, which may not occur in its absense. It would be very nice if the authors could show the SNARE complexes at the ER in the absence of BFA. There are possibly several possibilities to follow, but overexpression of SEC12 could for example retain the SNARE complex in a drug free manner.

---

## [Author Response]

*Essential revisions:*

*1) The possibility that SNARE molecules could assemble into the complexes during procedures of immunoprecipitation should be ruled out. It would be effective to add excessive amounts of purified proteins of or plant lysates containing KNOLLE and/or VAMP721 with specific tags into the reactions to show the exogenous proteins do not affect the amounts of the SNARE complexs.*

This criticism questions the general reliability of co-immunoprecipitation (co-IP) experiments, which are believed to reveal protein-protein interactions that occur in the living cell before its destruction.

In our study, we took care to reveal specific interactions by using two relevant controls:

1) KNOLLE forms 2 different SNARE complexes, as reported by us before (El Kasmi et al., 2013,): (i) KNOLLE, NPSN11, SYP71, VAMP721/722; (ii) KNOLLE, SNAP33, VAMP721/722. Furthermore, we have shown that those two complexes both contribute to cell plate formation. By immunoprecipitating the member of one complex (e.g. YFP-NPSN11) we co-immunoprecipitated other SNARE partners from the same KNOLLE complex (e.g. SYP71) but not a member of the other complex (e.g. SNAP33) and vice versa.

2) We also checked for the presence of the endosomal syntaxin (aka Qa-SNARE) PEP12 (aka SYP21) in the precipitate, again a negative result. Thus, we can exclude the possibility that our extraction procedure might disrupt selective protein-protein interactions and/or might lead to false positive signals because of insufficient solubilisation of the membranes on which the SNARE proteins reside. We hope that these controls are sufficient to demonstrate specificity of SNARE interaction in our co-IP experiments.

We would like to mention yet another evidence for the reliability of our co-IP procedure. We have reported that the linker sequence of KNOLLE is a binding site for the regulatory SEC1-like SM protein, KEULE, as shown by yeast two-hybrid assay and co-immunoprecipitation (Park et al., 2012). In that experiment, only KNOLLE was co-immunoprecipitated with HA-KEU but not tagged KNOLLE lacking its authentic linker sequence. Furthermore, the constitutively active form of KNOLLE was not co-immunoprecipitated with HA-KEU in the same experimental condition. These findings essentially rule out the possibility of non-specific assembly of SNARE complexes during protein preparation under our experimental conditions.

Due to limited material, we have used KN::mCherry:KNOLLE rather than KN::Myc-KN. We have performed the mixing experiment as proposed by adding to the experimental extract the same amount or 10x the amount of extract from *KN::mCherry-KN* transgenic seedlings before immunoprecipitation with anti-GFP beads. The results are shown in a new Figure 2—figure supplement 1 and mentioned in the text as follows: "In addition, to rule out that these complexes might have formed during the immunoprecipitation procedure, we mixed the protein extracts with varying amounts of extract from *KN::mCherry:KNOLLE* transgenic seedlings before immunoprecipitation with anti-GFP beads. Neither 1x nor 10x mCherry:KNOLLE addition changed the amount of KNOLLE-containing SNARE complex formed (Figure 2—figure supplement 1)".

*2) The immunoprecipitation experiments should be also accompanied by control using non-induced plants, Y:N11 and G:S33 plants without the EST treatment, which is useful to firmly show that authors detected only de-novo-synthesized proteins.*

We showed images of non-induced seedling root cells in which the KNOLLE signal was present at the cell plate but no signal of YFP-NPSN11 or GFP-SNAP33 was detectable (Figure 1).

We have performed the control experiments proposed, and the results of the immunoprecipitation analysis of non-induced seedlings are shown in a new panel to Figure 2—figure supplement 1 and referred to in the text as follows: "Further co-immunoprecipitation experiments with non-EST-induced seedlings demonstrated that co-immunoprecipitation of KNOLLE indeed required the EST-induced expression of YFP:NPSN11 or GFP:SNAP33 (Figure 2—figure supplement 1) (comment #2)." Because the co-IP experiment covers the Western blot analysis of total protein extract proposed above the latter has been omitted.

*3) It will be interesting to know if Keule is present in any of your GFP immune complexes, given its important role in cell plate formation.*

We have shown before that SEC1-like SM protein KEULE only interacts with monomeric KNOLLE, but not with the assembled KNOLLE complexes (using co-IP as well as pull-down from lysate with recombinant proteins). In addition, KEULE and KNOLLE only meet at the cell plate, but not prior to their arrival at that site (Park et al., 2012). When trafficking is blocked with BFA, KEULE still gets to the plane of division. Consistently, KEULE is not required for targeting of KNOLLE to the cell plate, and KNOLLE is not required for KEULE to get to the division plane (Waizenegger et al., Curr. Biol. 10, 1371; Park et al., 2012).

*4) The entire manuscript is based on drug treatments and it remains a possibility that BFA induces certain identity changes to the compartments or cytokinetic cells. Hence, the drug itself could actually stimulate complex assembly, which may not occur in its absense. It would be very nice if the authors could show the SNARE complexes at the ER in the absence of BFA. There are possibly several possibilities to follow, but overexpression of SEC12 could for example retain the SNARE complex in a drug free manner.*

We admit that there is a theoretical possibility that BFA might stimulate complex assembly. However, we consider this highly unlikely because of our several years of experience with BFA interference of membrane trafficking. The primary target of BFA has been identified as the SEC7 domain of ARF-GEFs in yeast and in Arabidopsis, and in both systems, BFA-sensitive ARF-GEFs have been rendered BFA-resistant. By rendering all three yeast ARF-GEFs (Gea1, Gea2, Sec7) BFA-resistant, yeast was enabled to grow on BFA-containing medium, indicating that there are no other essential BFA targets in yeast (Peyroche et al., 1999, Mol Cell 3, 275-285). We have also rendered BFA-sensitive ARF-GEFs BFA-resistant and vice versa by introducing single amino acid residue changes (M to L or L to M) (Geldner et al., 2003, Cell 112, 219-230; Richter et al., 2007; Richter et al., 2011, Nat Cell Biol 14, 80-86; Richter et al., 2014). In addition, the structure of the SEC7 domain complexed with the GDP-GTP exchange inhibitor BFA has been resolved (Mossessova et al., 2003; Renault et al., 2003). From a functional perspective, we have shown that BFA treatment resulting in BFA compartments can selectively effect either non-polar secretion of newly-synthesised auxin-efflux carrier PIN1 or the polar recycling of PIN1, depending on which of the endosomal ARF-GEFs, BIG3 or GNOM is (rendered) BFA-resistant (Richter et al., 2014). This specific example clearly demonstrates that the specific BFA-sensitive ARF-GEF(s) determines the physiological outcome of BFA treatment.

To address the possibility that BFA itself might stimulate SNARE complex assembly, we compared SNARE complex formation in BFA-treated with non-treated wild-type seedlings that expressed YFP-NPSN11 or GFP-SNAP33 by immunoprecipitation with anti-GFP beads. The results are shown as a new panel to Figure 2—figure supplement 1 and mentioned in the text as follows: "We also addressed whether BFA treatment might stimulate the formation of KNOLLE-containing SNARE complexes. To this end, we compared protein extracts from untreated wild-type seedlings with those from BFA-treated wild-type seedlings. No obvious difference in KNOLLE complex formation was detected between treated and untreated seedlings (Figure 2—figure supplement 1)".